# Efficacy and safety of tranexamic acid on blood loss and seizures in patients undergoing meningioma resection: A systematic review and meta-analysis

Xiaoyuan Liu[1‡], Minying Liu[1‡], Shu Li[1,2], Yue Ren[1], Maoyao Zheng[1], Min Zeng[1], Yuming Peng [1,2] *

1 Department of Anesthesiology, Beijing Tiantan Hospital, Capital Medical University, Beijing, China,
2 Outcomes Research Consortium, Cleveland, Ohio, United States of America

‡ XL and MS are contributed equally to this work and share first authorship.
* pengyuming@bjtth.org

## Abstract

### Introduction

It is unclear how tranexamic acid (TXA) affects blood loss and seizures in meningioma resections. We performed a systematic review and meta-analysis and tried to evaluate the effectiveness and safety of TXA use for patients undergoing meningioma resections.

### Methods

Regards to this systematic review and meta-analysis (registered with CRD42023416693), we searched PubMed, Embase (Ovid), EBSCO, and Cochrane central library up to and including Oct 2023. Patients undergoing meningioma resections treated with TXA and placebo or no treatment were eligible for this study. This would allow delineation of the impact of TXA on blood loss, postoperative seizure, and other complication incidences.

### Results

Four prospective cohort studies with 781 patients (390 patients in the TXA group and 391 patients in the control group) were conducted via a systematic review and meta-analysis. The results suggested that the application of TXA for patients undergoing meningioma resections reduced mean blood loss of 252 mL with 95% confidence interval (CI) -469.26 to -34.67 (P = 0.02) and $I^2$ of 94% but did not increase postoperative seizure (risk ratio: 1.08; 95%CI: 0.54 to 2.15; P = 0.84) and other complication rates.

### Conclusions

This systematic review and meta-analysis suggests that the administration of TXA could reduce blood loss in patients undergoing intracerebral meningioma resection.

**Data Availability Statement:** All relevant data are within the manuscript and its Supporting Information files.

**Funding:** This work was supported by the Beijing Municipal Administration of Hospitals Incubating Program (PX2022018), Beijing Municipal Administration of Seed Program (QMS20220518), and the Beijing Municipal Science & Technology Commission (Z191100006619068).The funding sponsor is neither involved in study design, collection, management, analysis, interpretation of data and report writing, nor the decision to submit the report for publication.

**Competing interests:** The authors have declared that no competing interests exist.

**Abbreviations:** TXA, tranexamic acid; PRISMA, Preferred Reporting Items for Systematic Reviews and Meta-analysis; WMD, weighted mean difference; RBC, red blood cell; CI, confidence interval; ICU, intensive care unit; Hb, hemoglobin.

## Registry information

The systematic review protocol has been registered at PROSPERO (Registration No. CRD42023416693) on April 23, 2023.

## Introduction

Meningiomas are one of the most common intracranial tumors and comprise approximately 36% of all intracranial neoplasms [1]. Although most meningiomas are benign [2], surgical removal is often accompanied by significant blood loss [3]. Excessive intraoperative blood loss may cause severe hemodynamic instability followed by cell salvage, large amounts of crystalloids, colloids, and allogeneic blood transfusion [4, 5].

TXA is a synthetical lysine analog [6–9] and plays its antifibrinolytic action by competitively inhibiting the binding of plasminogen to lysine residues on fibrin [6]. TXA could reduce blood loss and the need for transfusion in various surgical procedures, including trauma, orthopedics, otolaryngology, obstetrics, and cardiac surgery [10–17]. However, there remains a lack of consensus about the efficacy and safety of TXA administration in meningioma surgery.

Therefore, we performed a systematic review and meta-analysis, and tried to explore the potential evidence for the TXA's efficacy and safety in intracranial meningioma surgery.

## Methods

The systematic review protocol has been registered at PROSPERO (Registration No. CRD42023416693) on April 23, 2023. The systematic review and meta-analysis was carried out in accordance with the Preferred Reporting Items for Systematic Reviews and Meta-analysis (PRISMA) guidelines [18].

### Selection criteria

We incorporated trials that exhibited the following characteristics: (1) trial types: clinical randomized controlled trials/original papers, (2) population: patients undergoing meningioma resection, (3) intervention: at any dose of intravenous TXA, (4) control: placebo or without TXA administration, and (5) outcomes: the primary outcome was intraoperative blood loss, and transfusion requirements, postoperative seizure, and other complication rates are included as the secondary outcomes.

### Search strategy

To carry out a systematic review and meta-analysis about the administration of TXA in intracranial meningioma surgery, we searched PubMed, Embase (Ovid), EBSCO, and Cochrane central library up to and including Oct 15, 2023. Randomized controlled trials in the English language comparing TXA with placebo or no treatment in patients undergoing meningioma resection were eligible for this study. The keywords were employed to search PubMed with following mode: ((tranexamic acid) OR (TXA) OR (Tranexamic acid)) AND (meningioma) AND ((randomized controlled trial) OR (randomized)). The same search strategy for PubMed was used to retrieve another two databases [19].

### Data extraction

Articles identified by searching using the search terms were combined, and duplicates were removed. After the removal of duplicates, two independent reviewers (M L and S L) screened

titles, abstracts, and full-text articles for eligibility using pre-defined criteria. Any discrepancies or disagreements among reviewers were resolved by discussion. If disagreements occur clarification would be sought from the senior authors (YM P). This was illustrated using a PRISMA flow chart. Two authors (M L and S L) collected the study characteristics from each selected study, including publication time, the population, number of patients, the dose of TXA administered, intraoperative blood loss in mL, transfusion requirements, postoperative seizure within hospitalization, postoperative complications and information to assess the risk of bias in the studies.

## Quality assessments

M L and S L analyzed independently the risk of systematic errors (bias) of the trials incorporated in the meta-analysis based on the Cochrane Handbook, version 6.1. The Cochrane risk of bias tools were used to assess all the studies included. For randomized studies, the Risk of Bias version 2 (RoB 2) tool (version 22 August 2019) was used.

The assessment of bias risk was based on the following domains (1) randomization or non-randomization; (2) deviation from intended intervention; (3) missing outcome data; (4) measurement of the outcome; (5) selection of reported result.

## Outcomes

The primary outcome was intraoperative blood loss. The secondary outcomes were transfusion requirements and complication rate, including postoperative seizure, hematoma, and thrombosis events.

## Statistical analysis

Statistical analyses were conducted with Review Manager Software 5 (Review Manager [RevMan] Version 5.4 Copenhagen: The Nordic Cochrane Centre, The Cochrane Collaboration, 2020). For the continuous data, such as blood loss, the mean ± standard deviation was employed to calculate the weighted mean difference (WMD) and 95% confidence interval (CI) were applied. Statistical heterogeneity among the trials was tested by inspection of forest plot and calculation of $I^2$ statistics. A p-value < 0.1 and an $I^2$ value > 50% were considered to be suggestive of statistical heterogeneity. We utilized a random-effect model regardless of the $I^2$ caused by the meta-analysis application. All tests were two-tailed, and the result was presented as statistical significance when the P-value was less than 0.05.

## Results

### Literature search

Totally, we screened 326 abstracts, did a full-text review on 31 of them, and selected four trials enrolling 390 participants who received TXA usage and 391 participants with a placebo for the meta-analysis in this study (Fig 1).

### Description of selected studies

Table 1. summarizes the characteristics of selected trials. In three studies [3, 20, 21], TXA was administered through a loading dose of 20 mg/kg and a continuous infusion at 1 mg/kg/h until the completion of the surgery. Shu Li 2023 [19] involved a single dose of TXA at 20 mg/kg.

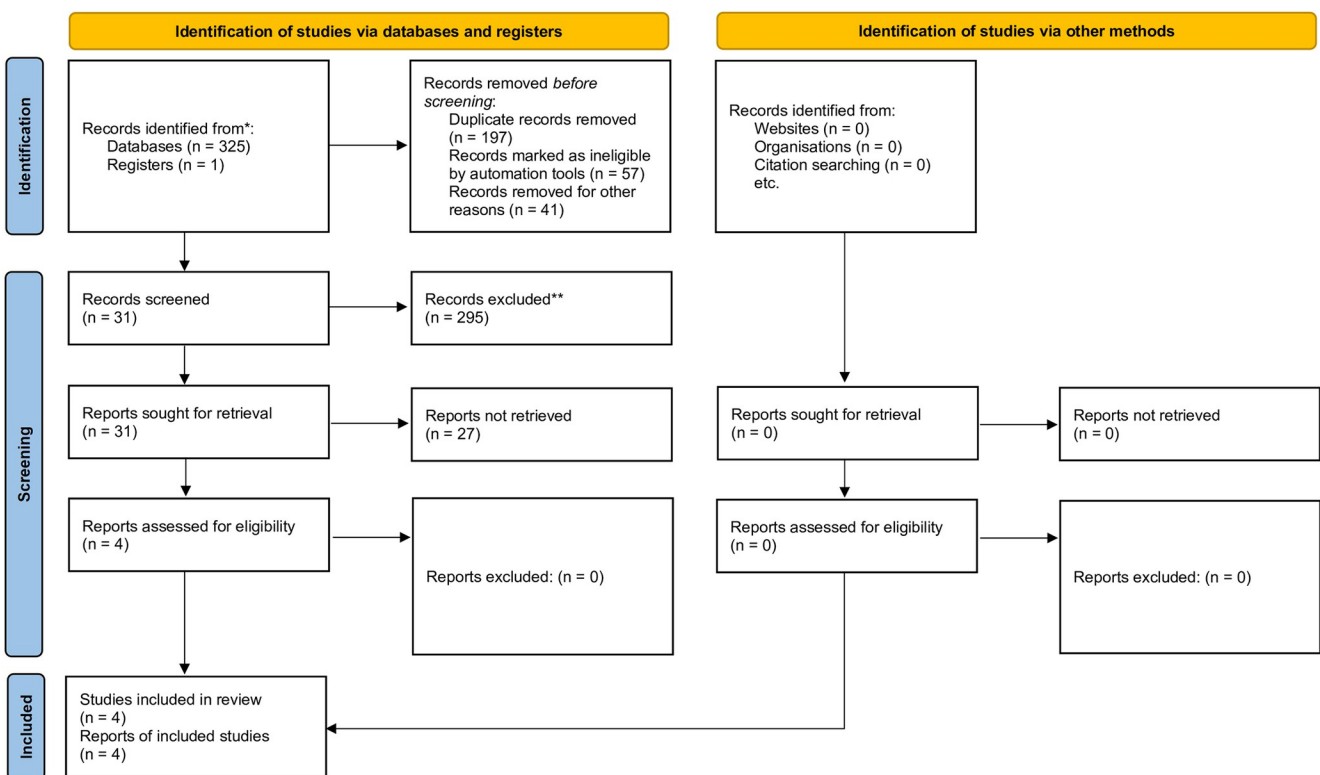

**Fig 1. PRISMA flowchart.**

### Risk of bias in individual trials

Overall, one trial [19] was regarded as low risk of bias. Four trials [3, 19–21] were deemed at low risk of bias for randomized process, deviation from intended interventions. One trial [19] was judged at low risk of bias for missing outcome data and selection of reported result. Supplementary material attached exhibits the risk of bias of the selected trials (S1 and S2 Figs).

### Outcomes

**Blood loss and blood transfusion requirements.** A total of 781 patients from four studies [3, 19–21] were eligible for the primary outcome. Compared with the control group, the administration of TXA distinctly reduced intraoperative blood loss of 252 mL with 95% CI -469.26 to -34.67 (P = 0.02; Fig 2A). A pooled WMD for all patients was completed, and the data showed significant heterogeneity ($I^2$ = 94%; Fig 2A).

Four studies [3, 19–21] with 781 patients reported the data on blood transfusion requirements. TXA did not reduce the probability of receiving red blood cell (RBC) transfusion (risk ratio: 0.75; 95%CI: 0.51 to 1.11; $I^2$ = 0%; P = 0.15; Fig 2A). Furthermore, the use of TXA did not decrease the amount of transfused RBC in two studies [3, 19] (mean difference: -46.59 mL; 95%CI: -107.70 to -14.52; $I^2$ = 56%; P = 0.14; Fig 2A).

**Incidence of postoperative seizure and incidence of other postoperative complications.** A total of 751 patients were included in three studies [3, 19, 21] relevant to this outcome. The use of TXA did not increase the risk of postoperative seizure compared with the

**Table 1. Summary characteristics of included trials.**

**A**

| Author, year | Total patients (n) | Design | Age (years) | Male (%) | TXA Dosing | Control | Duration of surgery(min) | Duration of anesthesia (min) | Blood loss (mL) | RBC transfusion (n) | FFP transfusion (n) | Cell saver (n) |
|---|---|---|---|---|---|---|---|---|---|---|---|---|
| Hooda et al., 2017 | 60 (30/30) | Prospective cohort study | 39.3 ±11.4 vs 41.6 ±11.2 | 46.67 vs 23.3 | 20 mg/kg 20 min followed by 1mg/kg/h | NS | 346±124 vs 359±129 | 436±122 vs 447±125 | 830 vs 1124 | 13 vs 17 | 4 vs 4 | 7 vs 6 |
| Rebai et al., 2021 | 91 (45/46) | Prospective cohort study | 49.5±8.7 vs 48.2 ±9.1 | 51 vs 56 | 20 mg/kg 20 min followed by 1mg/kg/h | NS | 220±31 vs 215±26 | 276±76 vs 302±86 | 283±71 vs 576±76 | 3 vs 4 | 0 vs 1 | - |
| Ravi et al., 2021 | 30 (15/15) | Prospective cohort study | 48.93 ±10.71 vs 52.0 ±13.18 | 33.3 vs 33.3 | 20 mg/kg followed by 1mg/kg/h | NS | 195±29 vs 212±24 | 217±24 vs 240±41 | 616±393 vs 1150±417 | 0 vs 6 | - | - |
| Shu Li et al., 2023 | 600 (300/300) | Prospective cohort study | 53±8 vs 53±8 | 31.7 vs 25 | Single dose of 20mg/kg | NS | 210±54 vs 210 ±60 | 270±54 vs 270±84 | 431±540 vs 423±416 | 14 vs 17 | 17 vs 17 | 51 vs 60 |

**B**

| Author, year | Total patients (n) | Design | Age (years) | Male (%) | TXA Dosing | Control | Postoperative seizure(n) | Hematoma (n) | Thrombosis events(n) | Re-exploration (n) | Antiepileptic drugs |
|---|---|---|---|---|---|---|---|---|---|---|---|
| Hooda et al., 2017 | 60(30/30) | Prospective cohort study | 39.3 ±11.4 vs 41.6 ±11.2 | 46.67 vs 23.3 | 20mg/kg 20min followed by 1mg/kg/h | NS | 1 vs 2 | 3 vs 4 | - | 1 vs 3 | None. |
| Rebai et al., 2021 | 91(45/46) | Prospective cohort study | 49.5±8.7 vs 48.2 ±9.1 | 51 vs 56 | 20mg/kg 20min followed by 1mg/kg/h | NS | 2 vs 2 | 2 vs 6 | 0 vs 0 | - | None. |
| Ravi et al., 2021 | 30(15/15) | Prospective cohort study | 48.93 ±10.71 vs 52.0 ±13.18 | 33.3 vs 33.3 | 20mg/kg followed by 1mg/kg/h | NS | - | 0 vs 0 | 0 vs 0 | - | None. |
| Shu Li et al., 2023 | 600(300/300) | Prospective cohort study | 53±8 vs 53±8 | 31.7 vs 25 | Single dose of 20mg/kg | NS | 13 vs 11 | 3 vs 5 | 26 vs 22 | 3 vs 6 | Yes.* |

Data is presented as experimental group versus (vs) control group respectively. FFP: fresh frozen plasma; RBC: red blood cell; TXA: tranexamic acid.

*Postoperative antiepileptic treatment includes intravenous or oral administration of valproic acid, oxcarbazepine, carbamazepine, levetiracetam, and phenobarbital.

Rescue intramuscular phenobarbital or continuous infusion of midazolam was administered when seizure fit attacks.

control group (risk ratio: 1.08; 95%CI: 0.54 to 2.15; $I^2$ = 0%; P = 0.84; Fig 2B). Only one study [19] mentioned the prophylactic administration of antiepileptic drugs postoperatively, while the other studies [3, 20, 21] did not specify this.

This systematic review and meta-analysis analyzed other postoperative complications, including hematoma, thrombosis events, and re-exploration. Data on postoperative hematoma were available for three studies [3, 19, 21], including 751 patients. The incidence of hematoma was not influenced by intraoperative use of TXA compared with controls (risk ratio: 0.55; 95% CI: 0.24 to 1.27; $I^2$ = 0%; P = 0.16; Fig 2B). A total of 48 thrombosis observations were extracted from one randomized controlled study [19]. The incidence of thrombosis events was also not influenced by TXA (risk ratio: 1.18; 95%CI: 0.69 to 2.04; P = 0.55; Fig 2B), whereas the incidence of re-exploration [3, 19] (risk ratio: 0.45; 95%CI: 0.14 to 1.43; P = 0.18; Fig 2B) was found similar between groups.

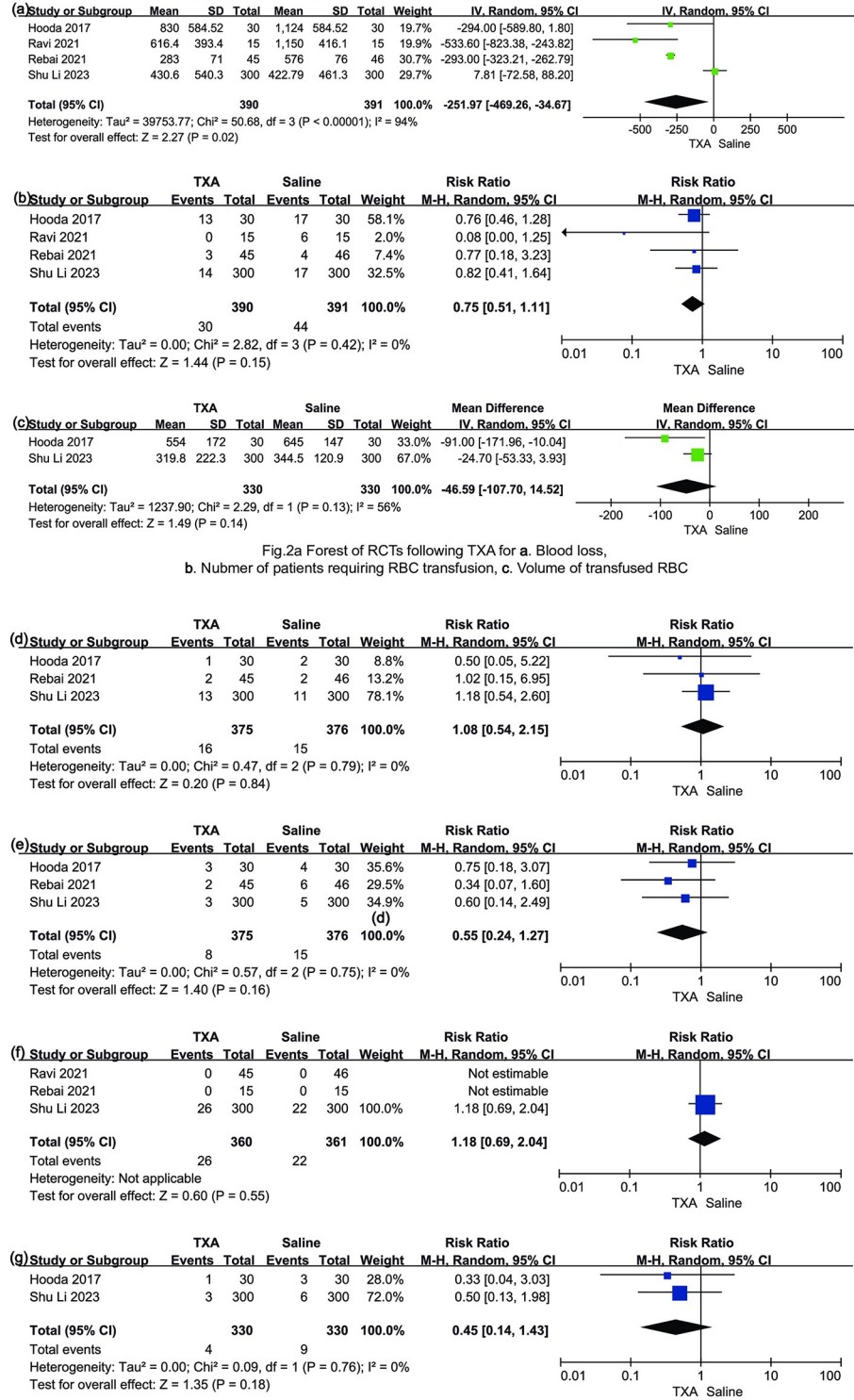

Fig.2a Forest of RCTs following TXA for **a**. Blood loss,
**b**. Nubmer of patients requiring RBC transfusion, **c**. Volume of transfused RBC

Fig.2b Forest of RCTs following TXA for postoperative complications **d**. Seizure,
**e**. Hematoma, **f**. Thrombosis events, **g**. Re-exploration

**Fig 2. a.** Forest of RCTs following TXA for a. Blood loss, b. Number of patients requiring RBC transfusion, c. Volume of transfused RBC. **b.** Forest of RCTs following TXA for postoperative complications d. Seizure, e. Hematoma, f. Thrombosis events, g. Re-exploration.

## Discussion

The systematic review and meta-analysis in our study showed that intravenous TXA administration could reduce blood loss in patients undergoing meningioma resections without increasing the risk of postoperative seizure or other postoperative complications.

Despite TXA having been reported to reduce blood loss and transfusion requirements in a variety of surgical settings, the use of TXA in patients undergoing neurosurgery is limited. A previous meta-analysis [22] conducted by Jeremiah et al. indicated administration of TXA reduced total blood loss of standardized mean difference of -1.40 (95%CI: -2.49 to -0.31; P = 0.01) and blood transfusion requirements risk ratio 0.58 (95%CI: 0.34 to 0.99; P = 0.48). Our results also showed that TXA could reduce a mean volume of 252 mL of intraoperative blood loss (95%CI: -469.26 to -34.67; P = 0.02; Fig 2A) with high heterogeneity ($I^2$ = 94%). We explored the following possible explanations for notable heterogeneity: first, differences exist in the populations enrolled across the four trials. Two trials [3, 20] enrolled patients with meningioma diameter $\geq$ 4 cm, while others [19, 21] had no criteria on tumor size. In general, the larger the tumor diameter, the more blood loss occurred during tumor resection. Furthermore, blood loss is influenced by tumor location, its relationship to venous sinuses, vascular supply, and whether there was perioperative embolization as well. And the estimation of blood loss by the anesthesiologist might be subject to a margin of error and differ in included trials. For blood transfusion requirements, transfusion trigger for RBC were also various in four trials. Three trials were hemoglobin concentration <8 g/dL [3, 19, 21], while another was hematocrit ≤27% [20]. The elements mentioned above might result in heterogeneity in transfusion requirements.

With regard to TXA dosing, official recommendations or guidelines are currently lacking. A prospective interventional dose-dependent study discovered that intraoperative low-dose TXA (10 mg/kg followed by 1 mg/kg/hour over 12 h) infusion effectively reduced intraoperative blood loss by 36% (365 mL vs. 552 mL; P = 0.042) in cardiac surgery [23]. A meta-analysis on the impact of a single preoperative dose of TXA found that administering a single dose of 10-20mg/kg TXA reduced 148 mL of blood loss and lower transfusion rates by 74% without a concomitant increase in the occurrence of venous thromboembolic events in orthopedic, obstetric, and gynecologic surgeries [24]. In a randomized controlled trial with 100 patients undergoing elective craniotomy for tumor excision, lower intraoperative blood loss was demonstrated (817 mL vs. 1084 mL; P = 0.012) in the TXA group with a bolus dose of 10–25 mg/kg followed by 1-10mg/kg/hour, while no patient in the study suffered from a complication related to the administration of TXA [25]. In our meta-analysis, the TXA dose of three trials [3, 20, 21] ranged from 4 g to 7 g, and Shu Li 2023 was approximately 100–150 mg [19]. None of the three studies [3, 20, 21] increased the incidence of TXA-associated perioperative complications. Moreover, our meta-analysis suggested that TXA reduced blood loss without increasing the risk of perioperative complications, which were in accordance with the findings of the individual study included in the meta-analysis. The meta-analysis [22] mentioned above also suggested that TXA did not result in a significant rise in the occurrence of postoperative seizure and hemostatic events.

### Limitations

Some limitations of our meta-analysis need to be considered. Firstly, the crucial data in a part of selected studies were missed, for example, a randomized controlled trial performed by Ravi et al did not provide the volume of transfused RBC and postoperative seizure [20]. In addition, we collected studies through systematically searching selected databases, nevertheless the potential for publication bias persists. Secondly, there was heterogeneity when comparing the

blood loss between the TXA and the control group. The multiple analyses in the studies chosen in the meta also led to the high value of $I^2$. The variables that may cause the heterogeneity include the difference in the dosing and timing of TXA administration, tumor size, tumor location, blood transfusion protocols, surgical techniques, approach and methods, as well as different strategies for measuring outcomes. Thirdly, only four studies were selected, and the sample size of the individual studies was limited. Additionally, there was no available data about thrombosis events for TXA administration in an anticoagulant state. Last, though two of the four studies excluded patients with preoperative seizures, only one study reported the use of anticonvulsive therapy, leading to an inconclusive effect of TXA on postoperative seizure.

## Conclusions

This systematic review and meta-analysis suggests that the administration of TXA could reduce blood loss in patients undergoing intracerebral meningioma resection. However, due to the heterogeneity of studies, the impact of TXA on the risk of seizure or other postoperative complications remains inconclusive. Large multi-center clinical trials are needed to fully confirm the effect of TXA administration in saving blood loss and postoperative seizures in the neurosurgical population.

## Supporting information

**S1 Checklist. PRISMA checklist.**
(DOCX)

**S1 Fig. Risk of bias item presented as percentages across all included studies.**
(TIF)

**S2 Fig. Risk of bias summary and graph.**
(TIF)

**S1 File. Minimal data set for meta-analysis.**
(ZIP)

## Acknowledgments

The authors gratefully thank the colleagues of the Neurosurgery department at Beijing Tiantan Hospital for their support and cooperation.

## Author Contributions

**Conceptualization:** Yuming Peng.

**Data curation:** Minying Liu, Shu Li, Yue Ren, Maoyao Zheng.

**Formal analysis:** Minying Liu.

**Funding acquisition:** Yuming Peng.

**Investigation:** Minying Liu, Yuming Peng.

**Methodology:** Shu Li, Yuming Peng.

**Project administration:** Shu Li.

**Supervision:** Shu Li, Min Zeng, Yuming Peng.

**Visualization:** Minying Liu, Yuming Peng.

**Writing – original draft:** Xiaoyuan Liu, Minying Liu.

**Writing – review & editing:** Xiaoyuan Liu, Minying Liu, Yuming Peng.

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
