## [Decision Letter · Decision Letter 0]

30 May 2024

PONE-D-24-02588Efficacy and safety of tranexamic acid on blood loss and seizures in patients undergoing meningioma resection: a systematic review and meta-analysisPLOS ONE

Dear Dr. Peng,

Thank you for submitting your manuscript to PLOS ONE. After careful consideration, we feel that it has merit but does not fully meet PLOS ONE’s publication criteria as it currently stands. Therefore, we invite you to submit a revised version of the manuscript that addresses the points raised during the review process.

We look forward to receiving your revised manuscript.

Kind regards,

Andreas K Demetriades, MBBChir, MPhil, FRCSEd, FEBNS.

Academic Editor

PLOS ONE

Journal Requirements:

3. Please include your tables as part of your main manuscript and remove the individual files. Please note that supplementary tables (should remain/ be uploaded) as separate ""supporting information"" files.

Additional Editor Comments:

Major review recommended

Reviewers' comments:

Reviewer's Responses to Questions

**Comments to the Author**

1. Is the manuscript technically sound, and do the data support the conclusions?

Reviewer #1: No

Reviewer #2: Yes

Reviewer #3: Yes

Reviewer #4: Yes

2. Has the statistical analysis been performed appropriately and rigorously? 

Reviewer #1: No

Reviewer #2: Yes

Reviewer #3: I Don't Know

Reviewer #4: Yes

3. Have the authors made all data underlying the findings in their manuscript fully available?

Reviewer #1: No

Reviewer #2: Yes

Reviewer #3: Yes

Reviewer #4: Yes

4. Is the manuscript presented in an intelligible fashion and written in standard English?

Reviewer #1: No

Reviewer #2: Yes

Reviewer #3: Yes

Reviewer #4: Yes

5. Review Comments to the Author

Reviewer #1: Dear Author,

The search strategy is not clear/available.

Literature suggests that Tranexamic acid can precipitate seizures in certain doses and there is controversy regarding the role of prevention of seizures.

It is not clear whether the patients in either group were on anti-epileptics.

Study needs to more elaborative.

Reviewer #2: Comments for Manuscript Submission ID PONE-D-24-02588

Comments for the Authors

I find it a well-designed paper. This study demonstrates that the administration of TXA could reduce blood loss (by 252 mL) in patients undergoing intracerebral meningioma resection. Meanwhile, TXA did not increase the risk of seizure, or other postoperative complications.

I, however, have a few minor comments to make:

Comment #1:

Meningiomas are one of the most common intracranial tumors

Line 232, on page 20, with null or hard-to-interpret outcomes may not be published; please, the author explicitly states the meaning of “hard-to-interpret outcomes.”

Comment #2:

Line 243-245, on page 21, This systematic review and meta-analysis suggests that the administration of TXA could reduce blood loss (by 252 mL) in patients undergoing intracerebral meningioma resection. The detailed blood loss (by 252) is inappropriate in the conclusion section.

Reviewer #3: Thank you for this valuable research. The manuscript is presented in a very good, informative and organised way. It is also written in a standard English language. Only the differences between the studies you choose for your analysis are multiple.

Reviewer #4: In their reponse letter, authors claim that recent study showed that TMZ did not add any benefit in the treatment of IDH wildtype glioblastoma compared with radiotherapy alone, regardless of MGMT promoter status (https://pubmed.ncbi.nlm.nih.gov/34000245/. This is simply a misinterpretation: Adjuvant temozolomide chemotherapy, but not concurrent temozolomide chemotherapy, was associated with a survival benefit in patients with 1p/19q non-co-deleted anaplastic gliomas. I suggest to change the passages in Discussion accordingly. I suggest to re-write the conclusion and state that the effects of the two drugs were shown in an in-vitro animal model under laboratory circumstances, and that it by no means suggest a positive clinical perspective for patients with necessity of further studies.

6. PLOS authors have the option to publish the peer review history of their article (what does this mean?). If published, this will include your full peer review and any attached files.

Reviewer #1: No

Reviewer #2: No

Reviewer #3: No

Reviewer #4: No

---

## [Author Response · Author response to Decision Letter 0]

2 Jul 2024

Response to Reviewers:

Reviewer #1:

The search strategy is not clear/available. Literature suggests that Tranexamic acid can precipitate seizures in certain doses and there is controversy regarding the role of prevention of seizures. It is not clear whether the patients in either group were on anti-epileptics. Study needs to more elaborative.

Response: Thanks. The search strategy for this paper was: ((tranexamic acid) OR (TXA) OR (Tranexamic acid)) AND (meningioma) AND ((randomized controlled trial) OR (randomized)). Tranexamic acid can precipitate seizures in certain doses, and we found that tranexamic acid did not increase the incidence of seizure in our meta-analysis. Additionally, the meta-analysis included four studies, of which only one mentioned the prophylactic administration of antiepileptic drugs postoperatively, while the remaining three did not specify whether antiepileptic treatment was given.

Author, year Whether to use postoperative antiepileptic drugs administration

Hooda et al., 2017 None.

Rebai et al., 2021 None.

Ravi et al., 2021 None.

Shu Li et al., 2023 Yes. Postoperative antiepileptic treatment includes intravenous or oral administration of valproic acid, oxcarbazepine, carbamazepine, levetiracetam, and phenobarbital. Rescue intramuscular phenobarbital or continuous infusion of midazolam was administered when seizure fit attacks.

Related revised manuscript text: Only one study mentioned the prophylactic administration of antiepileptic drugs postoperatively, while the other studies did not specify this.

Manuscript location: Page 9, line 19- and table 1 last column in the revised manuscript.

Reviewer #2: 

I find it a well-designed paper. This study demonstrates that the administration of TXA could reduce blood loss (by 252 mL) in patients undergoing intracerebral meningioma resection. Meanwhile, TXA did not increase the risk of seizure, or other postoperative complications. I, however, have a few minor comments to make:

Comment #1: Meningiomas are one of the most common intracranial tumors. Line 232, on page 20, with null or hard-to-interpret outcomes may not be published; please, the author explicitly states the meaning of “hard-to-interpret outcomes.”

Response: Thanks. We acknowledge that the term “hard-to-interpret” is inappropriate and have removed it.

Related revised manuscript text: In addition, we collected studies through systematically searching selected databases, nevertheless the potential for publication bias persists.

Manuscript location: Page 12, line 14- in the revised manuscript.

Comment #2:

Line 243-245, on page 21, This systematic review and meta-analysis suggests that the administration of TXA could reduce blood loss (by 252 mL) in patients undergoing intracerebral meningioma resection. The detailed blood loss (by 252) is inappropriate in the conclusion section.

Response: Thanks. We agree with you and have moved the data in the conclusion section.

Related revised manuscript text: This systematic review and meta-analysis suggests that the administration of TXA could reduce blood loss in patients undergoing intracerebral meningioma resection.

Manuscript location: Page 13, line 5- in the revised manuscript.

Reviewer #3:

Thank you for this valuable research. The manuscript is presented in a very good, informative and organised way. It is also written in a standard English language. Only the differences between the studies you choose for your analysis are multiple.

Response: Thanks. We appreciate your positive comments on the manuscript. We acknowledge that the multiple analysis in the studies chose in the meta which also led to the high value of I2. We also discussed this in detail in the limitations section to clarify the impact of these factors on our results.

Related revised manuscript text: The multiple analyses in the studies chose in the meta also led to the high value of I2.

Manuscript location: Page 12, line 17- in the revised manuscript.

Reviewer #4:

In their response letter, authors claim that recent study showed that TMZ did not add any benefit in the treatment of IDH wildtype glioblastoma compared with radiotherapy alone, regardless of MGMT promoter status (https://pubmed.ncbi.nlm.nih.gov/34000245/. This is simply a misinterpretation: Adjuvant temozolomide chemotherapy, but not concurrent temozolomide chemotherapy, was associated with a survival benefit in patients with 1p/19q non-co-deleted anaplastic gliomas. I suggest to change the passages in Discussion accordingly. I suggest to re-write the conclusion and state that the effects of the two drugs were shown in an in-vitro animal model under laboratory circumstances, and that it by no means suggest a positive clinical perspective for patients with necessity of further studies.

Response: Thanks. The reviewer’s comment appears to not correspond to our paper.

---

## [Decision Letter · Decision Letter 1]

7 Jul 2024

PONE-D-24-02588R1Efficacy and safety of tranexamic acid on blood loss and seizures in patients undergoing meningioma resection: a systematic review and meta-analysisPLOS ONE

Dear Dr. Peng,

Thank you for submitting your manuscript to PLOS ONE. After careful consideration, we feel that it has merit but does not fully meet PLOS ONE’s publication criteria as it currently stands. Therefore, we invite you to submit a revised version of the manuscript that addresses the points raised during the review process.

We look forward to receiving your revised manuscript.

Kind regards,

Andreas K Demetriades, MBBChir, MPhil, FRCSEd, FEBNS.

Academic Editor

PLOS ONE

Journal Requirements:

Reviewers' comments:

Reviewer's Responses to Questions

**Comments to the Author**

1. If the authors have adequately addressed your comments raised in a previous round of review and you feel that this manuscript is now acceptable for publication, you may indicate that here to bypass the “Comments to the Author” section, enter your conflict of interest statement in the “Confidential to Editor” section, and submit your "Accept" recommendation.

Reviewer #1: All comments have been addressed

Reviewer #4: All comments have been addressed

2. Is the manuscript technically sound, and do the data support the conclusions?

Reviewer #1: Yes

Reviewer #4: Yes

3. Has the statistical analysis been performed appropriately and rigorously? 

Reviewer #1: Yes

Reviewer #4: Yes

4. Have the authors made all data underlying the findings in their manuscript fully available?

Reviewer #1: Yes

Reviewer #4: Yes

5. Is the manuscript presented in an intelligible fashion and written in standard English?

Reviewer #1: Yes

Reviewer #4: Yes

6. Review Comments to the Author

Reviewer #1: The authors have made suggested changes.

Reviewer #4: Authors present a revised version of their manuscript -systematic review on role of transexamic acid in resection of meningiomas for prevention of postoperative bleeding as well as on prevention of seizures, where following anaylsis of four cohorts and more than 700 patients their main conclusion was that TXE reduces blood loss without increasing the risk of seizures. All limitations which were adressed, such as the fact that 3/4 cohort studies did not report use of anticonvulsive therapy - this needs to be adressed in the Conclusions and in Discussion.

7. PLOS authors have the option to publish the peer review history of their article (what does this mean?). If published, this will include your full peer review and any attached files.

Reviewer #1: No

Reviewer #4: No

---

## [Author Response · Author response to Decision Letter 1]

14 Jul 2024

Response to Reviewers:

Reviewer #4: Authors present a revised version of their manuscript -systematic review on role of transexamic acid in resection of meningiomas for prevention of postoperative bleeding as well as on prevention of seizures, where following anaylsis of four cohorts and more than 700 patients their main conclusion was that TXA reduces blood loss without increasing the risk of seizures. All limitations which were adressed, such as the fact that 3/4 cohort studies did not report use of anticonvulsive therapy - this needs to be adressed in the Conclusions and in Discussion.

Response: Thanks. We accepted your suggestion and added the limitation in the part of the conclusion and discussion. 

Related revised manuscript text: 

Last, though two of the four studies excluded patients with preoperative seizures, only one study reported the use of anticonvulsive therapy, leading to an inconclusive effect of TXA on postoperative seizure.

However, due to the heterogeneity of studies, the impact of TXA on the risk of seizure or other postoperative complications remains inconclusive. Large multi-center clinical trials are needed to fully confirm the effect of TXA administration in saving blood loss and postoperative seizures in the neurosurgical population.

Manuscript location: Page 13, line 6-8, line 13-16- in the revised manuscript.

---

## [Decision Letter · Decision Letter 2]

17 Jul 2024

Efficacy and safety of tranexamic acid on blood loss and seizures in patients undergoing meningioma resection: a systematic review and meta-analysis

PONE-D-24-02588R2

Dear authors

We’re pleased to inform you that your manuscript has been judged scientifically suitable for publication and will be formally accepted for publication once it meets all outstanding technical requirements.

Kind regards,

Andreas K Demetriades, MBBChir, MPhil, FRCSEd, FEBNS.

Academic Editor

PLOS ONE

Additional Editor Comments (optional):

Reviewers' comments:

Reviewer's Responses to Questions

**Comments to the Author**

1. If the authors have adequately addressed your comments raised in a previous round of review and you feel that this manuscript is now acceptable for publication, you may indicate that here to bypass the “Comments to the Author” section, enter your conflict of interest statement in the “Confidential to Editor” section, and submit your "Accept" recommendation.

Reviewer #4: All comments have been addressed

2. Is the manuscript technically sound, and do the data support the conclusions?

Reviewer #4: Yes

3. Has the statistical analysis been performed appropriately and rigorously? 

Reviewer #4: Yes

4. Have the authors made all data underlying the findings in their manuscript fully available?

Reviewer #4: Yes

5. Is the manuscript presented in an intelligible fashion and written in standard English?

Reviewer #4: Yes

6. Review Comments to the Author

Reviewer #4: Authors have sufficiently responded to reviewer remarks. I suggest this manuscirpt for publication.

7. PLOS authors have the option to publish the peer review history of their article (what does this mean?). If published, this will include your full peer review and any attached files.

Reviewer #4: No

---

## [Editor Report · Acceptance letter]

19 Jul 2024

PONE-D-24-02588R2 

PLOS ONE

Dear Dr. Peng, 

I'm pleased to inform you that your manuscript has been deemed suitable for publication in PLOS ONE. Congratulations! Your manuscript is now being handed over to our production team.

Kind regards, 

on behalf of

Dr. Andreas K Demetriades 

Academic Editor

PLOS ONE